



# Using coupled hydrodynamic biogeochemical models to predict the effects of tidal turbine arrays on phytoplankton dynamics

Pia Schuchert[1], Louise Kregting[1], Daniel Pritchard[2], Graham Savidge[3], Björn Elsäßer[1]

[1]School of Planning, Architecture and Civil Engineering, Queens University Belfast, Belfast, BT9 5AG, UK

[2]Department of Marine Science, University of Otago, PO Box 56, Dunedin 9054, New Zealand

[3]School of Biology, Queens University Belfast, Belfast, BT9 7BL, UK

*Correspondence to*: Louise Kregting (l.kregting@qub.ac.uk)

**Abstract.** Coupled 2-dimensional biogeochemical and hydrodynamic models offer the opportunity to predict potential effects of large scale tidal energy device (TED) arrays on the local and regional phytoplankton dynamics in coastal and inshore environments. In an idealised environment the effect of TEDs on phytoplankton dynamics accounted for up to 25% in phytoplankton concentrations, most likely associated with an increased residence time in an inshore basin. However, natural variation such as the intensity of photosynthetically active radiation had a larger effect on phytoplankton dynamics.

## 1 Introduction

Arrays of tidal energy devices (TEDs) have potential as a source of renewable energy, helping to assist in the effort to reduce carbon emission worldwide. They extract kinetic energy from the tides on a strongly cyclical and hence predictable basis and convert the extracted energy into electricity. Estimates suggest that large arrays of TEDs could provide a significant

proportion of the global electricity demand, with an estimated 32 GW in the UK alone (The Crown Estate 2012). However it is inevitable that they will have some effect on the ecosystem in the area in which they are deployed. To date there are only a few isolated TEDs in operation with various arrays in the planning stage in shallow (up to 50 m depth) coastal and inshore areas around the British Isles (The Crown Estate 2012), Canada (Cameron et al. 2015) and France (Magagna et. al. 2015). However in order to meet the 2050 targets of reducing carbon emissions by 80% of 1990s level, there are plans to deploy a

range of tidal arrays (The Crown Estate 2012).

There are several concerns about the likely environmental effect of large TED arrays along coastal inshore environments. A number of review studies discuss the potential local ecological effects (Boehlert and Gill 2010, Kadiri et al. 2012, Maclean et al. 2014, Shields et al. 2011, Shields and Payne 2014), focussing largely on sediment dynamics (Martin-Short et al. 2015,

Neill et al. 2009), collision risks with mammals (Thompson et al. 2013), fish or seabirds (Hammar and Ehnberg 2013, Hammar et al. 2015) and changes in larger community structures (Adams et al. 2014, Kregting et al. In review). The installation of a TED array will change the hydrodynamics of the ambient flow (Couch and Bryden 2007, Yang and Wang 2011). Many sessile and sedentary organisms depend on the flow of the water for availability of nutrients and food and hence changes in the hydrodynamics due to the presence of tidal turbines could potentially reduce the growth and ultimately

survival of these organisms (Shields et al. 2011). Hydrodynamic forces may also act to modify key predictors of phytoplankton derived primary production, either directly (e.g. transport of phytoplankton to deep water, thus reducing photosynthetically active radiation (PAR) available for photosynthesis) or indirectly (e.g. dilution and transport of nutrients, increased residence time in a basin) and therefore any changes to hydrodynamic conditions may have the potential to modify temporal and spatial patterns of primary production. Primary production supports higher trophic levels and therefore there is

a concern that anthropogenic effects which alter primary production and may multiply through the food chain in largely unknown ways.





In most marine ecosystems, the ultimate source of energy for primary production is the sun (photoautotrophic growth) with a key predictor of photoautotrophic growth being the availability of PAR (Falkowski and Raven 2007). However, nutrient availability is one of the limiting factors for phytoplankton growth in many marine environments such that in certain near-shore areas nutrient availability may be used as a key currency for primary production. High levels of nutrients (primarily

nitrates and ammonium) can cause eutrophication, leading to algal blooms and a range of symptomatic changes including deterioration of fisheries, biodiversity and water quality through reduction in dissolved oxygen and an ultimate decline in primary production associated with reduced penetration of PAR (Kadiri et al. 2012). Conversely, low nitrogen availability (oligotrophic conditions) can only sustain low levels of primary production, thus reducing energy available for higher trophic levels. The biochemical, biophysical and ecological processes that regulate phytoplankton derived primary production are

complex and a review of these processes is beyond the scope of this manuscript. Nevertheless, the global importance of phytoplankton ensures that the processes resulting in growth, mortality and nutrient remineralisation are included at a basic level in many numerical modelling studies of marine ecosystems.

As arrays of TEDs are still in the development stage, the best approach to determine the potential effects of their deployment

is to use numerical modelling approaches. Over the last 40 years, a simplified picture of controls on primary production and the interactions with other core components of the planktonic system has been widely studied using NPZD (Nutrient-Phytoplankton-Zooplankton-Detritus) models (Franks 2002). The model predicts changes over time in four state variables through a set of differential equations and have proven to be useful in capturing the properties and dynamics of a marine ecosystem at a higher level. The use of more complex models, i.e. models with greater number of state variables, has

inherent difficulties, including lack of data for model initialisation, verification and parameterisation (Ji et al. 2008), which may reduce the value of the model in its ability to answer key ecological questions (Hannah et al. 2010). NZPD models have been shown to be useful tools in predicting how an ecosystem is likely to change in response to changes in the physical and natural environment (Franks 2002). For example, the models have been used to show the consequences of intense aquaculture (Longdill 2007, Wild-Allen et al. 2010, Wild-Allen et al. 2011) and the impact of offshore wind farms (van der

Molen et al. 2014) and large scale (>100 km) impacts of TED array on primary production using the GETM-ERSEM-BFM model (van der Molen et al. 2016). However to date there are no studies that investigate the possible changes in primary production in close proximity to TED arrays.

This paper addresses the applicability of using coupled hydrodynamic and biogeochemical models to investigate near-field

(< 1 km), far-field (1–10 km) and regional scale (up to 30 km) effects of an array of tidal turbines on phytoplankton dynamics in a near coastal, shallow environment. In particular we use high resolution, 2-dimensional coupled hydrodynamic and biogeochemical NPZD model with two scenarios: no TEDs and an extreme, unrealistic setup with 55 TEDs, in an idealised domain.

**2 Materials and Methods**

**2.1 Hydrodynamic model**

Hydrodynamic and biogeochemical models were created using MIKE 21 software (DHI Water and Environment software package: www.dhisoftware.com). MIKE 21 FM is a two-dimensional, depth-averaged flexible mesh model based on a cell-centred finite volume method solution. For this study, an idealised, depth-averaged relatively shallow, model was used,

which was modified from the benchmark test case domain developed in (Kramer et al. 2014) with a tidal free surface forcing on the open boundary. The domain consisted of a high flow velocity channel between the shallow area of the open sea and an enclosed, shoaling out basin with some deeper channels of up to 50 m (Fig. 1). Grid cell sizes ranged from approximately 80 m$^2$ in the channel to 0.02 km$^2$ in the basin and between 0.02 and 4km$^2$ in the open sea. Temporal resolution was set to 15





minute time-steps for the output of hydrodynamic data and simulations were run over a whole year. Two different simulations were run, one with no turbines implemented into the domain and one including a setup of an array of 55 tidal turbines in the channel (Fig. 1). The MIKE 21 software models the effect of turbines on the hydrodynamics as sub-grid structures using a drag-law to capture the increasing resistance imposed by the turbine blades as the flow speed increases.

Simulated tidal turbines were based on the surface piercing horizontal axis tidal turbine SeaGen currently installed in the Strangford Narrows. The structure consists of a fixed cylindrical pile of 3 m diameter and 30 m height on which two separate 16 m diameter rotor blades on a large cross-arm are mounted. The centroid of the turbine was assumed to be in the middle of the water column.

To assure that the idealised model consisted of a well-mixed body of water, the "*h* over *U* cubed criterion" in Eq. (1)

$$\frac{h}{|U|^3} < 500 \qquad (1)$$

where $U$ is the mean tidal flow in m/s (i.e. the mean of maximum current speeds on both flood and ebb tides, ignoring direction) and $h$ the water depth in m, was used. It provides a rough predictor of the location of a summertime tidal mixing front separating zones with stratified and well-mixed water columns (Simpson and Hunter 1974, Thorpe 2007). In shallow regions of relatively fast tidal flows, as in the basin and the channel of the idealized model, the term is relatively small and

15 turbulence generated by shear stress on the bottom reaches the surface and results in mixing throughout the water column, sustaining the unstratified conditions (Thorpe 2007).

### 2.2 NPZD Model

A NPZD model following Fennel and Neumann (Fennel and Neumann 2015) was developed in MIKE DHI ECOLab. A 2-

20 dimensional model was chosen, omitting interactions with a sediment layer and sinking of phytoplankton and detritus. Nitrogen was used as the currency across the model (Fennel and Neumann 2015, Franks 2002). Only one generic type of phytoplankton and zooplankton were included, with process and growth rates loosely following Longdill (Longdill 2007)(Table 1). For each grid cell location, the time evolution of phytoplankton, nutrient, zooplankton and detritus concentrations is the sum of advection, diffusion and biogeochemical processes, which are described as:

(1) $\frac{dP}{dt} = \mathrm{P}growth - \mathrm{P}death - \mathrm{Z}graze - \mathrm{P}resp$

(2) $\frac{dZ}{dt} = \mathrm{Z}graze - \mathrm{Z}death - \mathrm{Z}excretion$

(3) $\frac{dD}{dt} = \mathrm{P}death + \mathrm{Z}death - \mathrm{D}mineralization$

(4) $\frac{dN}{dt} = \mathrm{P}resp + \mathrm{Z}excretion + \mathrm{D}mineralization - \mathrm{P}growth$

where P represents phytoplankton, N the growth limiting nutrient nitrogen consisting of pooled concentrations of $NO_3$ and $NH_4$, Z zooplankton and D detritus. A schematic model is shown in Fig. 2, showing the movement of nutrients from the nutrient pool through the different stages. The processes include light and nutrient dependent phytoplankton growth (*Pgrowth*), mortality (*Pdeath*), respiration (*Presp*) and grazing of zooplankton on phytoplankton (*Pgraze*) (Table 1). Other processes are mortality and excretion of zooplankton (*Zdeath* and *Zexcretion*) and mineralization of detritus back into the

nutrient pool (*Dmineralization*) (Table 1). The model includes key aspects of lower level trophic food web dynamics which are widely accepted in the marine ecosystem modelling community, such as Michaelis-Menten kinetics for phytoplankton nutrient uptake and zooplankton grazing and light-dependent growth of phytoplankton, i.e. photosynthesis. The intensity of PAR is a function of the surface PAR and the light attenuation profile averaged over the water depth. For the 2-dimensional hydrodynamic model, concentrations are averaged over the well-mixed water column. Biomass of phytoplankton,

zooplankton and detritus are described as total dry mass calculated using the Redfield-Ratio (Longdill 2007).





### 2.3 Data and Simulation

The NPZD model was coupled to each of the two hydrodynamic models: with and without an array of TEDs. For photoautotrophic growth four radiation (PAR) scenarios (A-D), based on four random years of PAR measurements made between 2004 and 2014 at the Queen's University Marine Laboratory in Portaferry, Northern Ireland, were conducted for
each of the coupled models. The PAR records were assumed to be representative of the strong natural annual variability in light conditions in North West Europe. The models were run over a one year period from October 1$^{st}$ to September 30$^{th}$. Prior to testing the scenarios a 4 year spin-up was run, one with each radiation scenario in a random order to assure a stable running of the system. Initial concentrations before the spin-up of nitrogen, phytoplankton, zooplankton and detritus were 5, 0.001, 0.001 and 5, without dimensions, respectively, following Fennel and Neumann (Fennel and Neumann 2015). In total
eight scenarios were conducted, each TED setup, without and with arrays, was run for each PAR (A-D) scenario.

### 2.4 Analysis

Changes in residence time in the basin and current speed were calculated from the hydrodynamic model setups with and without TEDs. In particular, due to the very basic shape of the basin, residence time $T$, number of tidal cycles, was calculated
as "flushing time", using the simple equation method Eq. (2)

$$T = \frac{V}{Q} \tag{2}$$

following (Herman et al. 2007), where $V$ is the mean volume of the basin and $Q$ the quantity of water which is exchanged during a tidal cycle.

To detect shifts in phytoplankton dynamics within the NPZD model only phytoplankton concentrations that were recorded every 15 min during the simulation from 25 sampling stations in the domain inside the basin, in the channel and the open sea were investigated (Fig. 1). While nutrient concentration, zooplankton and detritus were also recorded, these data are derived from a simulation based on differential equations and therefore dependent on each other and so results for phytoplankton are presented here. Daily average concentrations for each of the eight scenarios (two hydrodynamic settings with four PAR
scenarios each) were derived from the raw data. Additionally annual mean and peak/maximum concentrations were calculated. Visual interpretation and basic comparisons, such as time-series graphs, boxplots and differences between annual average mean and peak values were used in the first instance to investigate the effect of a TED array and any spatial or temporal variability. Linear regression and hierarchical partitioning following Groemping (2006) were conducted to quantify the effects and the relative importance of TEDs in comparison to spatial and temporal effects, however we omit significance
tests and p-values because these significance tests applied to simulated data are not particularly meaningful (White et al. 2014). All analyses were performed using R; hierarchical partitioning was implemented with the package *relaimpo* using the included LMG metric based on the work by Lindeman, Merenda and Gold (Groemping 2006, Lindeman et al. 1980).

### 3 Results

Average residence time of water in the basin increased by 5 %, from 6 days, 13 hours, 43 minutes without turbines to 6 days, 21 hours, 48 minutes with TEDs. Average differences in flow speed over one tidal cycle (12.4 hours) varied inside the channel (up to 0.32 m/s) but only to a small (< 0.04 m/s) amount in the basin and the open sea (Fig. 3). Water flow in the centre of the channel decreased with the introduction of the tidal turbines, while flow speed near the shore of the channel increased.

Time series of daily mean concentrations in the basin, channel and open sea (Fig. 4) showed stronger variation in phytoplankton concentration between PAR scenarios than TED scenarios. However, for each of the four PAR scenarios phytoplankton concentrations declined slightly faster and earlier when a TED array was present. Boxplots of mean





phytoplankton concentrations in the channel and basin also showed greater variation between PAR than TED scenarios (Fig. 5).

Comparisons between annual mean and peak (maximal) concentrations at each of the 25 sampling points of phytoplankton showed that differences in phytoplankton concentration as a result of natural annual radiation were considerably greater than those associated with the presence of the TED array (Table 2). The greatest difference in mean concentrations between PAR scenarios without TEDs was 4.42 g/m3 and with TEDs 3.72 g/m3, while the greatest observed difference between no TED/TED scenarios for PAR scenario A was 1.28 g/m3. Mean phytoplankton concentrations were generally 18-28% lower in scenarios with TEDs than without, except under PAR scenario C, in which phytoplankton concentration increased in the basin by 3.9%. Peak concentrations were between 0.3 and 13% lower with TEDs across all locations and PAR scenarios.

Linear regression and hierarchical partitioning models for annual mean and maximal concentrations of phytoplankton with PAR, TED scenario and location (basin, channel, open sea) as predictors showed that concentrations and dynamics varied between scenarios (Table 3). PAR was determined to be the most important factor in controlling phytoplankton dynamics, explaining 76% in the variation of mean phytoplankton concentrations but only 13% in variation of maximal peak concentration. TED absence/presence explained only 8% of mean phytoplankton concentration and 6% in variation of peak concentrations. The location, i.e. basin, channel or open sea, explained only 4% of variation in mean phytoplankton concentrations, but 7 % in maximal concentrations. In general, the predictors explained a total of 87 % of the differences in phytoplankton mean concentrations, but only 27% of the variation in maximum concentrations (Table 3).

## 4 Discussion

In the absence of any operational TED arrays, using coupled hydrodynamic with biogeochemical models provides the only approach to investigate possible changes on phytoplankton dynamics as a result of the installation of an array of TEDs. This is especially important at the planning stage. Using an idealised domain, in the near-field (50 m up to 5 km) and local scale (up to 30 km) the model predicted changes in phytoplankton dynamics as result of the changes in the hydrodynamics owing to the installation of a large array of TEDs. Even on this extreme case of energy extraction hydrodynamic influences on phytoplankton processes were however lower compared to the natural seasonal variation in changes in phytoplankton production. These results therefore suggest that in the absence of a full scale tidal energy array, coupled hydrodynamic and biogeochemical models provide the possibility to detect changes in phytoplankton dynamics as a result of changes in hydrodynamics.

The results suggest that natural variation in PAR could potentially have a greater impact on primary production than changes in hydrodynamics as a result of the installation of a very large TED array. Primary production generally shows natural annual variability which typically includes phytoplankton cyclical blooms, seasonal shifts and long term trends. The underlying mechanisms influencing this variability can vary both spatially and temporally (Cloern et al. 2005; Philippart et al. 2010) and depend on many interacting physical and biological factors, including light conditions, temperature, wind speed and species composition (Philippart et al. 2010). For example a long-term study in the North Sea showed peak chlorophyll *a* concentrations between 30 and 100 mg/m$^3$, a change of 300 % at the same spatial coordinates due to natural variation (Philippart et al. 2010). However, in addition to natural variation, anthropogenic changes can have a substantial impact on phytoplankton dynamics. Those include increased amounts of nutrients transported into the sea by rivers (such as fertilizer run-off from agriculture), sewage, aquaculture and mussel farming sites; the influence of such additions can be exacerbated through slower or less pronounced flushing of coastal areas or inlets after introduction of structures that slow down water or prevent movement. Thus, there is concern that large arrays of TEDs could have a substantial effect on the



flushing of a basin (Yang and Wang 2011). In the idealised model residence time increased by 5% under the TED array scenario. In response, a decrease in the average and cumulative phytoplankton concentrations by up to 20% was observed and phytoplankton concentrations decreased more quickly. In support of this phenomenon, enhanced grazing and growth time for the zooplankton population was observed. Indeed a slightly faster growth of the zooplankton concentration under

the TED scenario could be observed in the model (Fig. 6). However, in how far this effect could be observed in natural scenarios would depend on the actual species composition and parameters as they may behave differently to the generic forms chosen for this model.

This is a first attempt to investigate the effect of TED arrays on a local scale, using a high resolution 2D-model. While it

could be argued that 3-dimensional hydrodynamic models offer a range of additional facilities and the opportunity to include biological effects such as sinking and sediment processes, they also demand a high number of parameters. Often environments in which TED arrays are planned are poorly studied and the availability of ecological and biological information is scarce and inadequate. The parameterisation of a 3-dimensional model would hence be extremely difficult; the greater the number of unknown parameters the greater the uncertainties in the validation of the model and in the results.

Additionally, 3-dimensional models with a high spatial resolution as used in this study would lead to high computational costs. One reason to use of a 3-dimensional model would be if vertical stratification of the water column is likely, which can have major impacts on phytoplankton dynamics (Cloern et al. 2005). A decrease in flow speed through the introduction of a TED array might lead to stratified conditions in a previously unstratified area. In shallow (< 50m) coastal regions with relatively fast tidal flows, as in the idealized model of this study or the proposed areas for TED arrays (The Crown Estate

2012), the water column is well mixed supporting the unstratified conditions (Thorpe 2007). In case the environment to be modelled experiences seasonal stratification, or stratification due to freshwater input, this would have to be accounted for in a 3-dimensional model. However, in many areas where TED arrays are likely to be deployed, we consider that a 2-dimensional hydrodynamic model is appropriate and as we demonstrate here, this approach can provide significant insight without the added complexity of a fully 3-dimensional simulation.

As with any modelling approach, however complex, it is not possible to incorporate all predictors or processes of interest. Each model can only focus on a select number of processes, while others are omitted, combined or included indirectly. For example predation could generalised as mortality or sink term that removes matter from the system while omitting in-between processes. The modeller has to therefore critically assess the minimal necessary complexity of the model to find a

satisfactory solution for the problem that is being solved (Franks 2002). In this study only annual variation in PAR was included as a physical factor, while species composition was reduced to two generic categories of phytoplankton and zooplankton. Including additional state variables or processes would most likely mask the effect of a TED array and would make it more difficult to extract the degree to which the natural and anthropogenic factors contribute to the phytoplankton dynamics.

This study fills a currently missing link in the investigation of near field, coastal effects of TED arrays on primary production. While recent modelling studies on the possible near-field effects of tidal barrages (e.g. Severn Barrage) showed changes in the concentration of nitrogen and phytoplankton as well as changes in sediment transport the environmental impacts of a barrage structure cannot be directly compared to an array of individual turbines. A recent study by van der

Molen et al (2015) used a 3-dimensional modelling approach to predict the possible large scale effects of TED arrays in distances of 100s of kilometres. However this model had grid sizes in the range of kilometres rather than metres and therefore precludes the investigation of regional and near-field behaviour of phytoplankton dynamics.



In recent years much research into tidal turbine technology has focussed on the optimal design of the devices and the setup and optimisation of TED arrays (Culley et al. 2016, Stansby and Stallard 2016). These studies have shown that the setup constellation of an array, the number of turbines and relation to other devices will also effect on the hydrodynamics. In this study, the structure of the tidal turbine was based on the SeaGen turbine. It is to be expected the effect of an array of tidal

turbines might depend on the particular conditions of the site and community structure of organisms. Further the model in this study was parameterised with basic assumptions and concentrations obtained from published literature rather than direct measurements of a particular system, used only one single groupage of phytoplankton and zooplankton, and pooled all forms of nutrients into a single state variable. Despite these simplifications the changes in hydrodynamics as a result of an array of tidal turbines have been realistically modelled and it is likely that the approach used here would provide a good indication of

changes in NPZD dynamics if it were applied to a realistic array or parameterised with real (measured) values. All of those components will have to be investigated for any particular array to understand its specific impacts on the environment.

In conclusion this study showed that the approach is a valuable way in determining possible ecological effects of TED arrays on phytoplankton dynamics; a 2-dimensional approach offers the opportunity to use a high resolution grid while keeping the

15 computational costs and necessary data at a minimum. Our simulations show that, in this idealised system, TED arrays have an effect on primary production, however this is relatively small in comparison to natural variation. Further investigation is needed to implement a "living" system with realistic parameters and processes. This may include: additional zooplankton and phytoplankton species, a more complex channel with headlands and other features, filter feeders and other organisms settling on the structures and excess nutrient input entering the basin through agricultural run-offs or other sources. These

20 processes will be important in determining the realised effect of increased residence time in inshore loughs as a result of large TED arrays.

### 5. Acknowledgment

The research was financially supported by the EPSRC research grant EP/J010065/1.



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

**Figure 1: Idealised model of a channel, the outer border (on the left) and the basin (on the right). The inset displays the setup of the 55 TEDs in the channel. The depth in the channel and outside the channel is constant at 20m, while the basin has got some deeper areas of up to 60 m and is shoaling out towards the end. Concentrations of the state variables were recorded at the displayed 25 points, 1-9 in the basin, 10-16 in the channel and 17-25 outside the channel.**



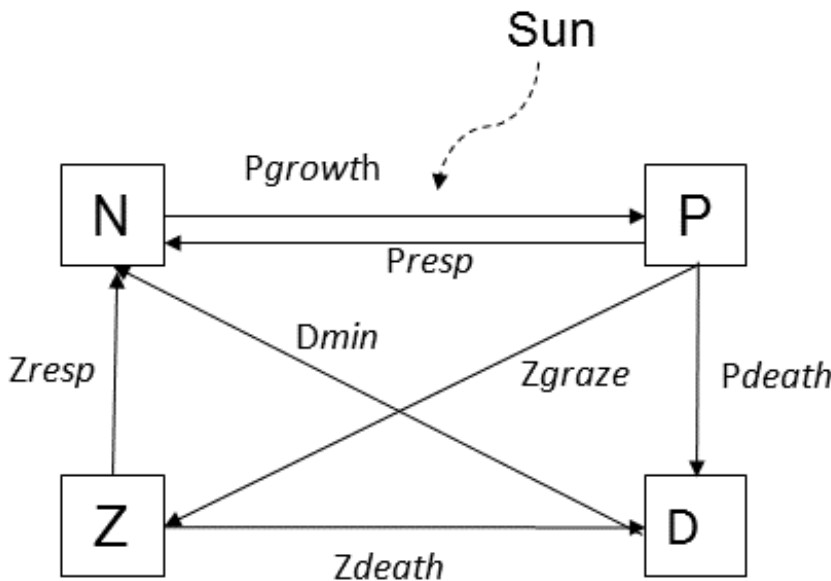

**Figure 2: Schematic NPZD model. State variables N (nutrients), P (phytoplankton), Z (zooplankton) and D (detritus) and the various processes affecting them.**




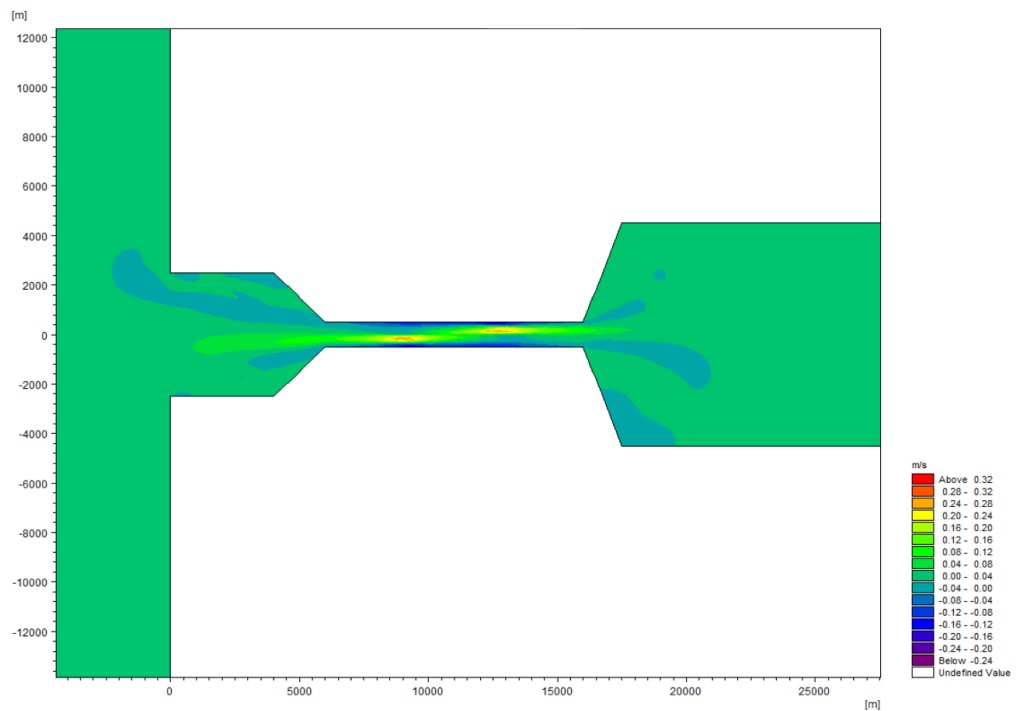

**Figure 3: Differences in current speed between scenarios with and without turbines, i.e difference of scenario without TED-Scenario with TED in m/s. The values are averaged across one tidal cycle (i.e. 12.4 hours). Outside the displayed area the**

25




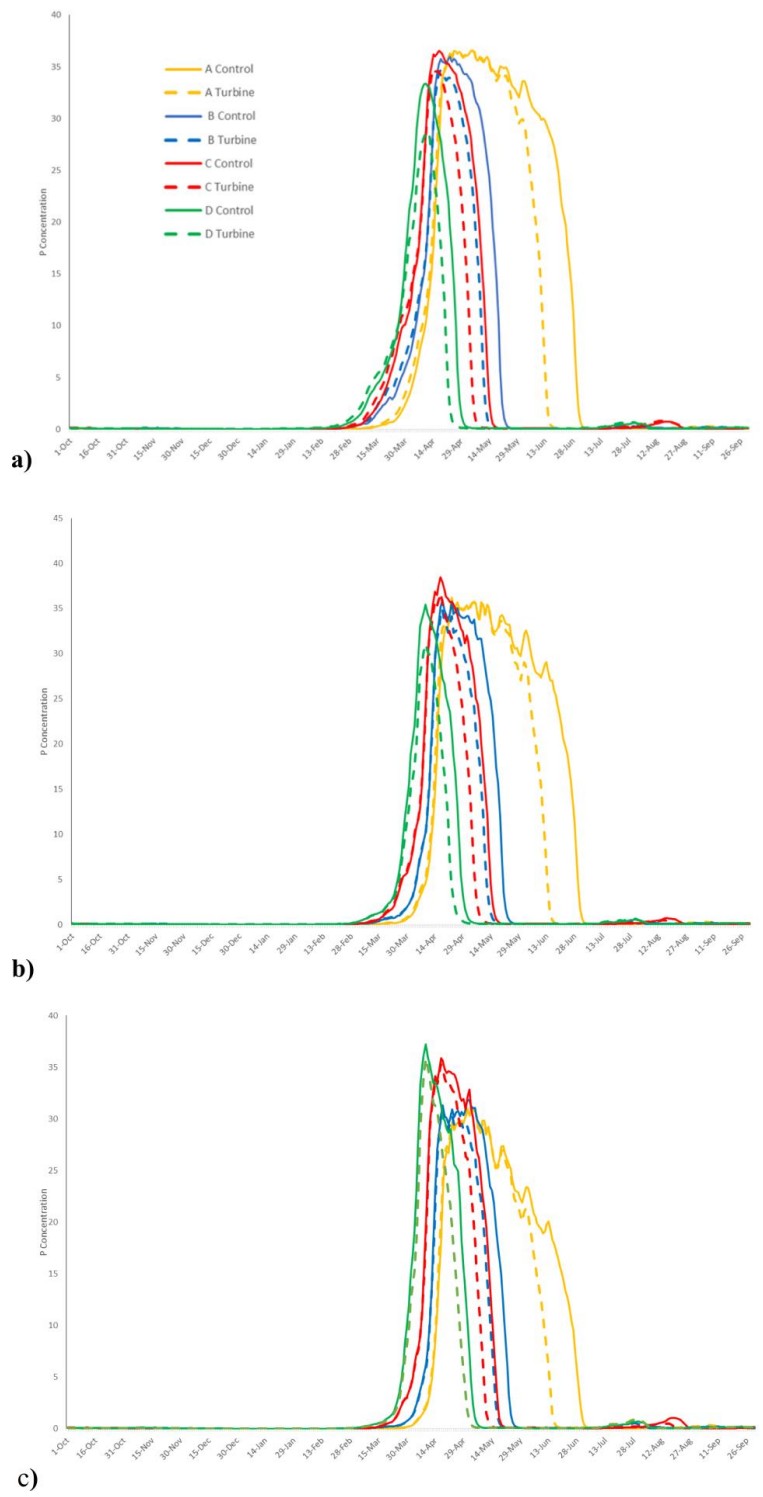

**Figure 4: Time-series of daily phytoplankton concentrations as average for the three different areas (Fig. 1), the basin (top), channel (middle) and open sea (bottom) for all four PAR scenarios A-D and scenarios with and without an array of TEDs.**



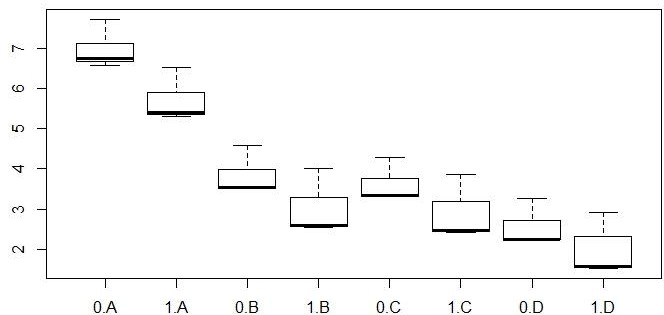

a)

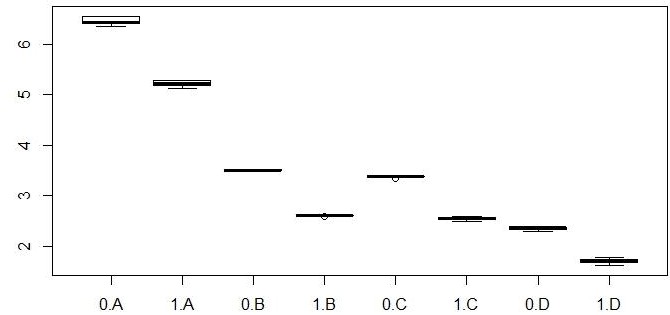

b)

**Figure 5: Boxplots for the mean concentrations of phytoplankton in the basin (a), in the channel (b). A-D refer to the PAR scenarios and 0 and 1 to the scenarios without (0) and with (1) an array of TEDs. 0. A hence refers to PAR scenario A without turbines.**



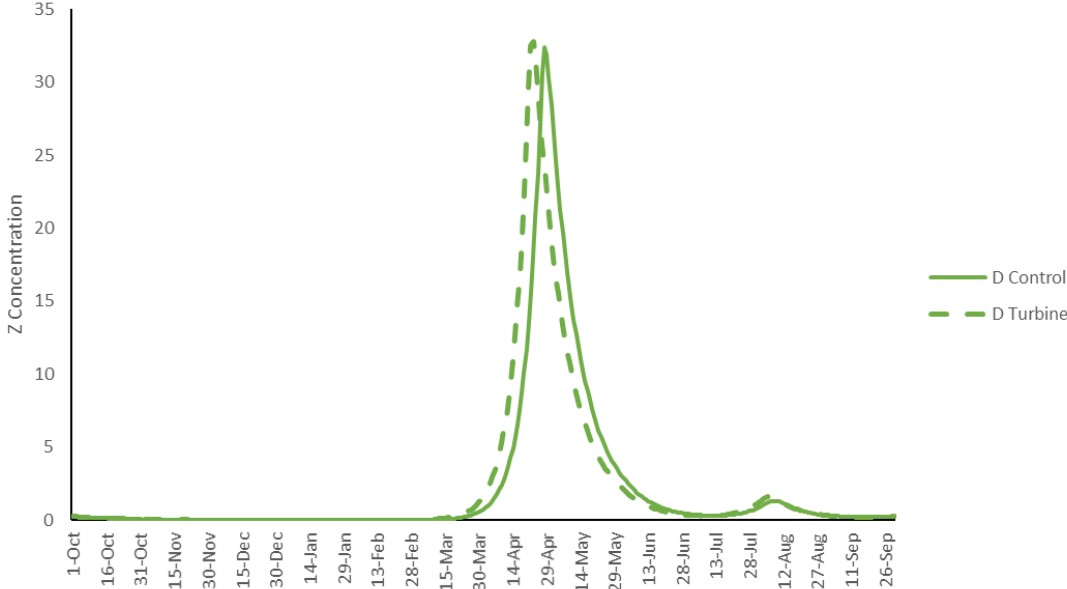

**Figure 6: Annual variation in zooplankton concentration in the basin under PAR scenario D.**



**Table 1: State variables, psrocesses, rates and constants of the 2D NPZD model.**

| Name | Type | Process | Description |
|---|---|---|---|
| N | State | $-Pgrowth+Prespiration+Dmineralization+Zexcretion$ | Nutrient concentration |
| P | State | $Pgrowth-Prespiration-Pdeath-Pgraze$ | Phytoplankton concentration |
| Z | State | $Pgraze-Zdeath-Zexcretion$ | Zooplankton Concentration |
| D | State | $Pdeath-Dmineralization+Zdeath$ | Detritus Concentration |
| Prespiration | aux | lpn*P | Phytoplankton respiration |
| Pdeath | aux | lpd*P | Phytoplankton mortality |
| Pgraze | aux | gP*Z*Epz | Grazing of Zooplankton on Phytoplankton |
| Pgrowth | aux | Rmaxa*fN*P | Growth of Phytoplankton |
| Zdeath | aux | lzd*Z | Zooplankton mortality |
| Zexcretion | aux | lzn*Z | Zooplankton excretion |
| Dmineralization | aux | ldn*D | Detritus mineralization |
| fI | aux | $\frac{I}{Iopt} * \exp(1 - \frac{I}{Iopt})$ | Light limiting function for phytoplankton growth |
| I | aux | $Max(0.00001, \frac{\frac{Ios}{dz}*(1-\exp(-eta*dz))*1}{eta})$ | Average light intensity I from the surface to the depth dz. Lambert-Beer expression has to be integrated over depth. |
| gP | aux | Gmax*fP | Zooplankton grazing rate |
| fP | aux | $IF(P>Pt), THEN \frac{P-Pt}{(Kp+P-Pt)}, ELSE\ 0$ | Phytoplankton limitation function |
| Rmaxa | aux | Rmax*fI | Maximum growth rate of phytoplankton light dependent |
| fN | aux | $\frac{N}{Kn + N}$ | Nutrient limitation function |
| Kn | const | 0.025 | Half saturation constant |
| lpn | const | 0.1 | Phytoplankton respiration |
| lpd | const | 0.001 | Phytoplankton mortality rate |
| ldn | const | 0.005 | Detritus mineralization rate |
| gmax | const | 0.4 | Maximum grazing rate of zooplankton |
| lzd | const | 0.05 | Zooplankton mortality rate |
| lzn | const | 0.035 | Zooplankton excretion rate |
| rmax | const | 1 | Phytoplankton maximal growth rate |
| eta | const | 0.34 | Light attenuation factor in water column |
| Epz | const | 0.6 | Feeding efficiency of zooplankton |
| Pt | const | 0.04 | Phytoplankton Threshold for zooplankton feeding |
| Kp | const | 0.2 | Half Saturation constant for Phytoplankton |
| dz | forcing | | Depth |
| ios | forcing | | PAR |





**Table 2: Annual mean and peak phytoplankton concentrations for each scenario, averaged over the three locations: Basin, Channel and Open Sea (Fig. 1). Maximal differences between concentrations in PAR scenarios and between TED scenarios.**

| | Radiation | | Basin | Channel | Open ocean |
|---|---|---|---|---|---|
| **Without TED** | A | Average concentration | 6.94 | 6.46 | 4.87 |
| | | Peak concentration | 37.32 | 36.19 | 31.13 |
| | B | Average concentration | 3.81 | 3.51 | 3.11 |
| | | Peak concentration | 36.81 | 35.92 | 32.62 |
| | C | Average concentration | 3.59 | 3.38 | 3.25 |
| | | Peak concentration | 37.15 | 38.48 | 36.07 |
| | D | Average concentration | 2.52 | 2.35 | 2.60 |
| | | Peak concentration | 33.60 | 35.43 | 37.23 |
| | Largest difference between scenarios A-D | Average concentration | 4.42 | 4.11 | 2.27 |
| | | Peak concentration | 3.72 | 3.05 | 5.92 |
| **With TED** | A | Average concentration | 5.65 | 5.23 | 4.08 |
| | | Peak concentration | 37.13 | 36.07 | 31.28 |
| | B | Average concentration | 2.95 | 2.61 | 2.47 |
| | | Peak concentration | 35.42 | 34.98 | 31.96 |
| | C | Average concentration | 3.73 | 2.55 | 2.62 |
| | | Peak concentration | 33.02 | 36.71 | 35.31 |
| | D | Average annual concentration | 1.94 | 1.70 | 2.09 |
| | | Peak concentration | 29.27 | 31.03 | 35.85 |
| | Largest difference between scenarios A-D | Average concentration | 3.72 | 3.52 | 1.99 |
| | | Peak concentration | 7.85 | 5.67 | 4.57 |
| **Difference in raw value and percentage between scenarios No TEDs and TEDs; (TED- No TED)/No TED** | A | Average concentration | -1.28 (-18.6%) | -1.23 (-19%) | -0.79 (-16.2%) |
| | | Peak concentration | -0.19 (-0.5%) | -0.12 (-0.3%) | -0.04 (-0.5%) |
| | B | Average concentration | -0.85 (-22.6%) | -0.89 (-25.6%) | -0.63 (-20.6%) |
| | | Peak concentration | -1.39 (-3.8%) | -0.93 (-2.6%) | -0.66 (-2.0%) |
| | C | Average concentration | 0.14 (3.9%) | -0.83 (-24.6%) | -0.63 (-19.4%) |
| | | Peak concentration | -4.13 (-11.1%) | -1.77 (-4.6%) | -0.76 (-2.1%) |





| | D | Average concentration | -0.58 (-23.0%) | -0.65 (-27.7%) | -0.51 (-19.6%) |
|---|---|---|---|---|---|
| | | Peak concentration | -4.32 (-12.9%) | -4.39 (-12.4%) | -1.38 (-3.7%) |





Table 3: Results of General linear models for mean, max and cumulative values of phytoplankton and maximal and cumulative values of nitrogen. The variables turbine, location and radiation year are categorical variables, with the following categories: Turbine : no Turbine/ Turbine; Location: Basin/Channel/Open Ocean, and Radiation: A/B/C/D. Residual Standard Error, Multiple R-Squared, F and p-value are not displayed as they are not meaningful in this context (White et al. 2014)

| | | | Variable | Estimate | Std. Error | Variation explained (%) |
|---|---|---|---|---|---|---|
| P mean Total Variation explained: 87.14% | | | Intercept | 6.23 | 0.09 | |
| | TED | | Turbine | -0.79 | 0.07 | 7.8 |
| | Area | | Channel | -0.31 | 0.09 | 3.6 |
| | | | Open Ocean | -0.64 | 0.08 | |
| | PAR | | B | -2.44 | 0.10 | 75.6 |
| | | | C | -2.47 | 0.10 | |
| | | | D | -3.30 | 0.10 | |
| P max Total Variation explained: 27.24% | | | Intercept | 35.88 | 0.43 | |
| | TED | | Turbine | -1.46 | 0.33 | 7.4 |
| | Area | | Channel | 0.36 | 0.41 | 7.0 |
| | | | Open Ocean | -1.29 | 0.39 | |
| | PAR | | B | -0.23 | 0.46 | 12.8 |
| | | | C | 1.63 | 0.46 | |
| | | | D | -1.01 | 0.46 | |

