# Peer review of "Using coupled hydrodynamic biogeochemical models to predict the effects of tidal turbine arrays on phytoplankton dynamics"

_Biogeosciences, 2016_

## Referee Comment (RC1) · Anonymous Referee #1 · 6 Jul 2016

This paper aims to investigate the local scale (< 30 km) impact of tidal turbine (TED) arrays on phytoplankton dynamics (i.e. mean and peak concentrations) in coastal and inshore areas around the British Isles. An unstructured 2D hydrodynamic model coupled with a simple NPZD model was used to simulate this impact in an idealized domain. The results suggest that, in an idealized environment, the presence of TED arrays leads to a slight increase in water residence time followed by a decline in phytoplankton concentrations; however, such impact is much smaller in comparison to other environmental variables that are subject to substantial natural variability, such as PAR as demonstrated by the model. This is one of the first attempts to investigate the local scale impact of large TED arrays on lower trophic level food web dynamics, and I

agree that the results presented in this manuscript will ultimately shed light on water resource management and guide policy decision making. However, in order to satisfy the interests of BG readers, a good number of concerns must be addressed in a revised manuscript before publication can be considered. Some of my quibbles (major points and specific comments) are listed below.

**Major Points:**

1. The importance of nutrients and light to phytoplankton growth has been well known. It is also convincing that natural variability of these variables can be substantial, therefore scenario experiments should be conducted (4 PAR scenarios  $\times$  2 TED setups) in order to understand the relative impact of TED arrays. However, it does not seem to justify that the full range of natural variability in PAR can be sufficiently represented by 'random' measurements made between 2004-2014. Which four years were selected? Are those data year-round, time-series measurements? How do they look like, and how does seasonal variability compare to interannual variability? Did the authors choose the four PAR scenarios to represent high, medium, and low light conditions? The authors should provide these observational data in the manuscript and discuss this further in depth. I would also recommend the authors to analyze remotely sensed observations (e.g. 4 km MODIS) to assess the long-term variability of PAR in that area.

Along the same line, why did the authors not include nutrient scenarios in their numerical experiments? The authors mentioned that anthropogenic eutrophication may result in serious environmental issues in coastal regions. While the model has such capacity (e.g. by changing the initial nutrient concentration), the impact of eutrophication on phytoplankton is not being addressed in this study.

2. I understand that it is not necessary to describe the detailed setup of the hydrodynamic model as it follows a previously published study. The authors mentioned that the hydrodynamic impact of TED arrays is presented as sub-grid structures using a drag-law; however, I suggest the authors should at least describe the equation of drag force as well as other model parameters (e.g. turbine thrust coefficient) they used in this study. Also, the resulting modifications in current velocities and flushing time might not play a major role in driving phytoplankton growth. The simplifications made in the 2D model simulations (i.e. omitting vertical variability and benthic flux) is in the meantime ignoring physical processes that are critical to phytoplankton. For example, turbid turbine wakes (up to 150 m in width and several km in length) have been frequently observed from satellite images (e.g. Vanhellemont and Ruddick, 2014). These wakes not only increase light attenuation thereby reducing light availability, but also enhance nutrient supply to phytoplankton. These caveats should be further discussed in the manuscript.

3. The authors chose to run two model simulations, one with no turbines and one with an extremely large TED arrays (55 turbine). How was this number (i.e. 55) chosen? The hydrodynamic model suggests that the effect of TED arrays on flushing time is very small (5%). Are the modeled results sensitive to the setup of TED arrays at all (e.g. turbine location; number of tidal turbines; turbine parameterization)? Is 5% within model error? The authors should perform tests to determine the sensitivity of the hydrodynamic model results and the robustness of the conclusions on the effect of TED arrays on hydrodynamic processes.

4. The linkage between the deployment of TED arrays and the modeled decline in phytoplankton concentrations was poorly discussed in the manuscript. I would assume that nutrient flux across the channel is reduced with the presence of TED arrays, resulting in a decrease in phytoplankton growth. It is surprising and unclear to me why zooplankton grazing rate would be enhanced instead.

Minor Points:

1. '... following Fennel and Neumann (Fennel and Neumann 2015)' should be '...following Fennel and Neumann (2015)'

2. P.3, I. 25-28: Eq.(1)-(4) should be Eq.(2)-(5).

3. P3 I. 30-40: I suggest the authors tighten up the wording here and move some of the model descriptions into the caption of Fig.2.

4. P.4, I. 8-9: "Initial concentration..., without dimensions, respectively". Here the authors claimed that the 4 variables used in this study are non-dimensional. However, at P.5, I. 7-9, the authors added unit for the phytoplankton concentration, please clarify or eliminate.

5. P.4 l.16: Eq. (2) should be Eq.(6).

6. P.4 I. 22-24: The authors write "While nutrient concentrations,.....": I do not agree with this statement. Indeed the variables in NPZD model are dependent on each other. However, the results for each variables also have unique phenomenon. Fig.4a and Fig.6 gave good examples – a lag between phytoplankton production and grazing by zooplankton. Also the other variables could be useful to explain the reason why phytoplankton concentrations decrease due to the deployment of TED arrays. I suggest this sentence be removed and the authors should provide more modeled results.

7. P.4, I. 30: Lindeman, Merenda and Gold (Groemping 2006, Lindeman et al. 1980); what is the reference for Lindeman, Merenda and Gold? Is Groemping 2006 and Lindeman et al. 1980 referencing Lindeman, Merenda and Gold?

8. P.6, I. 37-38: Please add reference for this sentence.

9. P.6, I. 39-40: "van der Molen et al (2015)" should be "van der Molen et al (2016)".

10. P.14, Fig.4: Should use same upper bound of y-axis for all the three figures.

11. P.15, Fig.5: The caption of Fig5 is not complete. Please check.

12. P.16, Fig.6: Why was only zooplankton for Scenario D shown? What about A-C?

13. In the sections of results and discussion, only the averaged (over several sampling locations) phytoplankton concentrations have been shown for the three different areas (e.g. Fig.5; Table 2). However, spatial variation is expected within each area (especially

СЗ

inside the basin). Such variation should be presented as standard deviation in each table cell.

14. P.20: Table 3: The authors write "Results of General linear models for mean, max and cumulative values of phytoplankton and maximal and cumulative value of nitrogen." This sentence is not consistent with the text (P.5 I. 12-14) "Linear regression and hierarchical partitioning for annual mean and maximal concentration of phytoplankton with .....". Please check and clarify.

In summary, the authors present a potential minor effect of tidal turbine (TED) arrays on phytoplankton concentrations in an idealized environment using a coupled hydrodynamic and biogeochemical model. This manuscript might be potentially of interest to the BG readers. However, the current version of manuscript is structured more like a report. The description of the coupled model and the discussion of model results are superficial and the authors should spend additional effort to improve it. The authors should also pay attention to the formatting of references in the text. I recommend the editor reject this manuscript, but permit a resubmission upon a major revision that addresses the concerns listed above.

References:

Q. Vanhellemont, K. Ruddick, Turbid wakes associated with offshore wind turbines observed with Landsat 8 Remote Sens. Environ.,145 (2014), pp. 105–115

---

## Referee Comment (RC2) · Anonymous Referee #2 · 8 Jul 2016

This manuscript presents a modeling study of hydrodynamics and biogeochemistry, aimed at evaluating the effects of increased drag (due to turbines) within a tidal inlet on phytoplankton dynamics. The paper is well-written and relatively clear, though lacking in detail throughout. Given the highly idealized nature of the study, I do not think the results are generally applicable to other systems, or provide any specific insight that couldn't be otherwise attained through a simple "thought" experiment.

For example, the primary result, i.e. residence time increased and had a measurable effect on phytoplankton concentrations, is specific to this idealized model domain. The magnitude of this change is directly a function of the inlet and basin configuration and turbine density. If the study evaluated multiple instances of configurations and

densities, and generated a more widely applicable evaluation of residence time (and phytoplankton concentrations), then I believe this would be of broader interest.

The use of a 2D model, though defended in the discussion, is a major shortcoming. Given the depths in the model, it is unclear to me how the vertical profile of light, primary production, and phytoplankton concentration can be properly resolved. The assumption of a well-mixed water column, through a h/u3 criterion, is never actually quantified anywhere in the paper. And regardless of this criterion, one would expect significant vertical structure during neap tides when mixing is reduced.

The specification of drag force is never clearly explained. Representing momentum extraction and turbulence dissipation by structures is complex and needs to be explained in more detail. Again, given the depth of the channel, representing the three-dimensional structure of the hydrodynamics may be important.

The offshore, inlet, and basin configuration used here seem relatively uncommon. I cannot think of realistic systems with such a configuration. If the authors believe this to be some prototypical system, then more justification should be given.

Specific comments:

Abstract: more detail is needed here, regarding findings and implications.

Section 2.1: more detail needed on model setup, drag parameterization, forcing conditions. Mixing criterion is described here, but not evaluated anywhere else in the paper.

Section 2.2: more detail needed for PAR input specification. How is PAR averaged over such large depths to yield a realistic value for phytoplankton growth?

Section 2.3: Why is a 4-year spin up needed, if concentrations appear to fall to zero in the winter?

Section 2.4: This quantification for residence time is relatively primitive and should be compared with values obtained through particle tracking, dye release, or other means.

Again, vertical structure of hydrodynamics will be important in a system with these depths.

Section 3: There is almost no detail given for hydrodynamic results. What is the spatial structure of velocity? What is the spatial structure of the mixing criterion on spring and neap tides? When would the assumption of well-mixed conditions fail? How would that affect the implications?

СЗ

---

## Referee Comment (RC3) · Anonymous Referee #3 · 12 Jul 2016

General Comments

This study conducted a series idealized numerical experiments to explore the effects of tidal turbine arrays on phytoplankton dynamics. The topic is of interest. However, I feel that this manuscript needs to be considerably fleshed out, and, therefore, would not recommend publication in biogeosciences at current status.

Specific comments

The setup of idealized models is always highly simplified. This simplification, however, must have a realistic reference. Otherwise, the idealized model would be just a toy. This study designed a model domain that consists of a tidal inlet and a semi-circle coastal

sea. Does this domain represent the typical region around the British Isles where TEDs will be deployed? Is the setup of TEDs in the inlet channel the standard design of future TED deployment? If they are not, the representativeness of this idealized numerical model is less meaningful.

As idealized models are idealized, there is no need to validate the model results. Nonetheless, the model results should "look" generally reasonable. The simulated phytoplankton concentration is almost zero during the whole year in addition to a spring bloom occurred in April. Is this a typical phytoplankton cycle around the British Isles? Specially, what is the atmospheric forcing of the model? What are the seasonal variations of the PAR scenarios? Did the model simulate the variation of water temperature that is important to phytoplankton growth?

The main founding of this study is that the deployment of TEDs reduces the phytoplankton concentration on basis of the comparison of the model results with/without TEDs. Certainly, this comparison is useful, but looks too superficial. Of importance is to explore the key processes (either physical or biological) through which the deployment of TEDs alters the phytoplankton dynamics. I feel that this manuscript would be more valuable if in-depth analysis of those processes were presented.

---

## Author Comment (AC3)

6th September 2016

Dear Associate Editor,

Re: **Using coupled hydrodynamic biogeochemical models to predict the effects of tidal turbine arrays on phytoplankton dynamics** authored by P. Schuchert et al. MS No.: bg-2016-232

We would like to thank the reviewers for their time in reading this manuscript and for their constructive suggestions and comments. We believe their comments will substantially improve the manuscript. In our detailed response below, we have *paraphrased the reviewers comment in italics*, then provided our response in blue.

**Anonymous Referee #1**

Major Points:
*1. The importance of nutrients and light to phytoplankton growth has been well known. It is also convincing that natural variability of these variables can be substantial, therefore scenario experiments should be conducted (4 PAR scenarios _ 2 TED setups) in order to understand the relative impact of TED arrays. However, it does not seem to justify that the full range of natural variability in PAR can be sufficiently represented by 'random' measurements made between 2004-2014. Which four years were selected? Are those data year-round, time-series measurements? How do they look like, and how does seasonal variability compare to interannual variability? Did the authors choose the four PAR scenarios to represent high, medium, and low light conditions? The authors should provide these observational data in the manuscript and discuss this further in depth. I would also recommend the authors to analyze remotely sensed observations (e.g. 4 km MODIS) to assess the long-term variability of PAR in that area.*
We agree that we require more additional detail explaining the PAR data. The use of the word 'random' measurements is perhaps misleading as the data are constructed from real data from the Queen's University Marine Laboratory (QML) and could be better described as a 'synthetic' dataset constructed from real data at QML. Some data were constructed owing to there being three full years of PAR data available, but the additional fourth year had weeks missing and so was interpolated in order to get a full data set. In addition we will include a figure of the PAR data in order to visually observe the seasonal and inter-annual variability. We don't believe that we require the need to investigate the MODIS data as well. We used four different years that spanned from 2004 to 2014 which is relatively long-term variability and don't believe remotely sensed observations would add any overall value to our results.

*Along the same line, why did the authors not include nutrient scenarios in their numerical experiments? The authors mentioned that anthropogenic eutrophication may result in serious environmental issues in coastal regions. While the model has such capacity (e.g. by changing the initial nutrient concentration), the impact of eutrophication on phytoplankton is not being addressed in this study.*
This is another good point raised by the reviewer. Essentially this is a domain based on Strangford Lough (R2 mentions this too and we will address that this domain is based on Strangford Lough). We therefore based our initial nutrient concentrations on what we observe in Strangford Lough. From unpublished observations, eutrophication does not appear to have been an issue in Strangford Lough over a period of at least 30 years or more owing to the small catchment area of the Lough and flushing time. We agree that the effects of eutrophication on phytoplankton growth are, of course, also important, but this is an explorative study and we can add a section in the discussion addressing that these scenarios may also need to be explored, depending on the area investigated.

*2. I understand that it is not necessary to describe the detailed setup of the hydrodynamic model as it follows a previously published study. The authors mentioned that the hydrodynamic impact of TED arrays is presented as sub-grid structures using a drag-law; however, I suggest the authors should at least describe the equation of drag force as well as other model parameters (e.g. turbine thrust coefficient) they used in this study. Also, the resulting modifications in current velocities and flushing time might not play a major role in driving phytoplankton growth. The simplifications made in the 2D model simulations (i.e. omitting vertical variability and benthic flux) is in the meantime ignoring physical processes that are critical to phytoplankton. For example, turbid turbine wakes (up to 150 m in width and several km in length) have been frequently observed from satellite images (e.g. Vanhellemont and Ruddick, 2014). These wakes not only increase light attenuation thereby reducing light availability, but also enhance nutrient supply to phytoplankton. These caveats should be further discussed in the manuscript.*

Yes, we will add more detail in the methods on the implementation of the turbines in the hydrodynamic model. We agree with the reviewer that with the introduction of a tidal energy structure, at the near-field scale, will alter the local hydrodynamics. While turbine wakes are observable, we do have measurements of turbulence on the seabed near the SeaGen structure which showed no increase in turbulence. We can only assume from the lack of increased turbulence adjacent to the seabed by the SeaGen infrastructure that turbidity levels will not be enhanced. Further, turbidity is unlikely to be an issue given the boulder/rocky nature of the seabed in most locations that tidal turbines will be implemented, however we still require information to show this. We will as the reviewer suggests add this information into the discussion.

*3. The authors chose to run two model simulations, one with no turbines and one with an extremely large TED arrays (55 turbine). How was this number (i.e. 55) chosen?*

The large array of 55 turbines was chosen because it was classed as high (perhaps, unreasonably high) number of turbines in the channel. It was 55 specifically (and not 54 or 56) because the initial idea was to remove "blocks" of 5 turbines and simulate 0, 1, 5…. 55. We therefore have two arrays of 25, plus the "block" of 5 in the middle (as shown in Figure 1). The reasons for this specific layout (and not just a single long "block") was to provide some reality to the layout of the array by maintaining a navigation channel through the centre of the array. This is information we can add into the methods section.

*The hydrodynamic model suggests that the effect of TED arrays on flushing time is very small (5%). Are the modeled results sensitive to the setup of TED arrays at all (e.g. turbine location; number of tidal turbines; turbine parameterization)? Is 5% within model error? The authors should perform tests to determine the sensitivity of the hydrodynamic model results and the robustness of the conclusions on the effect of TED arrays on hydrodynamic processes.*

The implementation of the turbines as a drag coefficient / stress (energy) sink is explicit and thus extracts energy at the given rate. The question of course is if the chosen coefficient resembles the actual energy extracted close enough. This would vary from device to device, but in principle it can be 'made' correct (through calibration/validation, model scale experiments or full scale measurements). The point is that under feasible conditions in reality only a limited amount of energy will be extracted by tidal stream turbines which may only be 5 or 10% of the resource. The objective of in stream tidal energy versus tidal barrages or similar is that it extracts less energy at an even lower cost, thus maximising the yield while not harming the environment. Coming back to the numerical simulations, whether or not a tidal model can be accurate to 5% in its energy estimate depends on the quality of the model. Realistically we doubt that a lot of models achieve this, and we are looking at differences between modelling scenarios which are most certainly within the modelling assumptions valid. We can add something along those lines to the manuscript.

*4. The linkage between the deployment of TED arrays and the modeled decline in phytoplankton concentrations was poorly discussed in the manuscript. I would assume that nutrient flux across the channel is reduced with the presence of TED arrays, resulting in a decrease in phytoplankton growth. It is surprising and unclear to me why zooplankton grazing rate would be enhanced instead.*

We agree with the reviewer and will clarify this aspect of the discussion to address the reviewer's comments. It is possible that the earlier onset of zooplankton growth might be the reason for a faster decline in phytoplankton. However this is not necessarily the reason and other possible reasons will be addressed in the discussion. Different zooplankton or phytoplankton species may be reacting differently to the changes in the environment, but in our scenarios, the zooplankton and phytoplankton species are generic.

Minor Points:

*1. ': : : following Fennel and Neumann (Fennel and Neumann 2015)' should be*
*': : :following Fennel and Neumann (2015)'*

We will address.

*2. P.3, l. 25-28: Eq.(1)-(4) should be Eq.(2)-(5).*

We will address.

*3. P3 l. 30-40: I suggest the authors tighten up the wording here and move some of the model descriptions into the caption of Fig.2.*

We will address.

*4. P.4, l. 8-9: "Initial concentration: : :., without dimensions, respectively". Here the authors claimed that the 4 variables used in this study are non-dimensional. However, at P.5, l. 7-9, the authors added unit for the phytoplankton concentration, please clarify or eliminate.*

We will address.

*5. P.4 l.16: Eq. (2) should be Eq.(6).*

We will address.

*6. P.4 l. 22-24: The authors write "While nutrient concentrations,: : :: : :": I do not agree with this statement. Indeed the variables in NPZD model are dependent on each other.*

It is possible the reviewer mis-read this as we do say that the differential equations are 'dependent on each other'.

*However, the results for each variables also have unique phenomenon. Fig.4a and Fig.6 gave good examples – a lag between phytoplankton production and grazing by zooplankton. Also the other variables could be useful to explain the reason why phytoplankton concentrations decrease due to the deployment of TED arrays. I suggest this sentence be removed and the authors should provide more modeled results.*

Agree with the reviewer and will tighten up the discussion on page 6.

*7. P.4, I. 30: Lindeman, Merenda and Gold (Groemping 2006, Lindeman et al. 1980); what is the reference for Lindeman, Merenda and Gold? Is Groemping 2006 and Lindeman et al. 1980 referencing Lindeman, Merenda and Gold?*

We will address.

*8. P.6, l. 37-38: Please add reference for this sentence.*

Will do.

*9. P.6, l. 39-40: "van der Molen et al (2015)" should be "van der Molen et al (2016)".*
Will check.

*10. P.14, Fig.4: Should use same upper bound of y-axis for all the three figures.*
Will alter.

*11. P.15, Fig.5: The caption of Fig5 is not complete. Please check.*
Will check.

*12. P.16, Fig.6: Why was only zooplankton for Scenario D shown? What about A-C?*
We did not want to focus on zooplankton. Scenario D is shown for discussion reasons only, making a possible explanation more visual. The other figures can be included, but we believe this will not provide any added value to the manuscript. As noted earlier, the four state parameters are dependent on each other.

*13. In the sections of results and discussion, only the averaged (over several sampling locations) phytoplankton concentrations have been shown for the three different areas (e.g. Fig.5; Table 2). However, spatial variation is expected within each area (especially inside the basin). Such variation should be presented as standard deviation in each table cell.*
We can include this information. It was left out due to the size of the table and making it more difficult to read. Alternatively we could include the full table (i.e. each location) into an appendix.

*14. P.20: Table 3: The authors write "Results of General linear models for mean, max and cumulative values of phytoplankton and maximal and cumulative value of nitrogen." This sentence is not consistent with the text (P.5 l. 12-14) "Linear regression and hierarchical partitioning for annual mean and maximal concentration of phytoplankton with : : :: : :". Please check and clarify.*
Will check.

*In summary, the authors present a potential minor effect of tidal turbine (TED) arrays on phytoplankton concentrations in an idealized environment using a coupled hydrodynamic and biogeochemical model. This manuscript might be potentially of interest to the BG readers. However, the current version of manuscript is structured more like a report. The description of the coupled model and the discussion of model results are superficial and the authors should spend additional effort to improve it. The authors should also pay attention to the formatting of references in the text. I recommend the editor reject this manuscript, but permit a resubmission upon a major revision that addresses the concerns listed above.*
As the reviewer suggests we would address their concerns as listed above and ensure that we have checked the formatting of the references and any other formatting issues.

**Anonymous Referee #2**

*Given the highly idealized nature of the study, I do not think the results are generally applicable to other systems, or provide any specific insight that couldn't be otherwise attained through a simple "thought" experiment. For example, the primary result, i.e. residence time increased and had a measurable effect on phytoplankton concentrations, is specific to this idealized model domain. The magnitude of this change is directly a function of the inlet and basin configuration and turbine density. If the study evaluated multiple instances of configurations and densities, and generated a more widely applicable evaluation of residence time (and phytoplankton concentrations), then I believe this would be of broader interest.*

It is correct that for this simplified model the results could also be derived from a simple thought experiment. However for a more complex domain with, for example, variable bathymetries, this may not be the case. The purpose of the paper is that one can derive these results through modelling, where the thought experiment would become untenable due to the large number of interacting effects in the hydrodynamics alone. Also it is correct to note that residence time in any case will be a very good first measure as to what will happen, and checking that the simulation are similar to some 'back of the envelope' estimates is a basic rule for every numerical modeller. Again we can add a comment to this effect to the manuscript.

*The use of a 2D model, though defended in the discussion, is a major shortcoming. Given the depths in the model, it is unclear to me how the vertical profile of light, primary production, and phytoplankton concentration can be properly resolved. The assumption of a well-mixed water column, through a $h/u^3$ criterion, is never actually quantified anywhere in the paper. And regardless of this criterion, one would expect significant vertical structure during neap tides when mixing is reduced. The specification of drag force is never clearly explained. Representing momentum extraction and turbulence dissipation by structures is complex and needs to be explained in more detail. Again, given the depth of the channel, representing the threedimensional structure of the hydrodynamics may be important.*

In order to clarify how Ecolab resolves the vertical profile of light, primary production and phytoplankton, more detail will be added to the methods section. This information is in the (publically available) manual of Ecolab and we will therefore cite this document. In regards to the values for the $h/U^3$ criterion, we have calculated the values throughout the domain. The assumption of a well-mixed water column holds everywhere except the very shallow areas at the head of the inside the lough. This information can be added. In regards to the drag force and turbulence, these concerns are similar to those of Reviewer #1 concerns and believe by addressing the comments made by Reviewer #1 we also address the concern of Reviewer #2.

*The offshore, inlet, and basin configuration used here seem relatively uncommon. I cannot think of realistic systems with such a configuration. If the authors believe this to be some prototypical system, then more justification should be given.*

We can understand the concern of the reviewer, but it is an idealised domain with realistic tidal velocities and the maximum number of turbines in the channel and accords with the site of the world's first grid-compliant TED (i.e. Strangford Lough, as discussed in response to Reviewer #1). While the domain is similar to Strangford Lough, the domain was also based on previously published domains such as by Walkington & Burrows (2009) which we will cite. The results suggest that light will be more influential in phytoplankton growth than an alteration of hydrodynamics. The results were not just for in the bay, but out in the open sea as well, so, yes, this system is realistic in the sense you have harbours/inlets throughout the world or high flowing channels that lead out into areas of lower flow rates. We can provide more detail to this in the use of the domain.

Walkington, I., & Burrows, R. (2009). Modelling tidal stream power potential. Applied Ocean Research, 31(4), 239-245.

**Specific comments:**

*Abstract: more detail is needed here, regarding findings and implications.*
As the reviewer suggests we will add more detail.

*Section 2.1: more detail needed on model setup, drag parameterization, forcing conditions. Mixing criterion is described here, but not evaluated anywhere else in the paper.*
Yes we can add more detail on the model setup and include the mixing criterion.

*Section 2.2: more detail needed for PAR input specification. How is PAR averaged over such large depths to yield a realistic value for phytoplankton growth?*
As detailed in our response earlier to this reviewer, we will add more detail on this section in the methods.

*Section 2.3: Why is a 4-year spin up needed, if concentrations appear to fall to zero in the winter?*
This is a valid point and will clarify this information in the methods.

*Section 2.4: This quantification for residence time is relatively primitive and should be compared with values obtained through particle tracking, dye release, or other means. Again, vertical structure of hydrodynamics will be important in a system with these depths.*
We believe that the quantification for residence time is a valid method in areas without headlands and will clarify this in the manuscript as well as cite the manuscript that shows this (Yang and Wang 2011).

*Section 3: There is almost no detail given for hydrodynamic results. What is the spatial structure of velocity? What is the spatial structure of the mixing criterion on spring and neap tides? When would the assumption of well-mixed conditions fail? How would that affect the implications?*
We believe Figure 3 in the manuscript shows nicely the differences in current speed between the scenarios with and without turbines. As mentioned previously we can add the values of the $h/U^3$ results for the various sampling points with and without turbines and will discuss in greater detail regarding stratification. We also state in the discussion, and can further add, how a change in mixing conditions could change the outcome although we did not observe a change in mixing conditions caused by the deployment of the turbines.

**Anonymous Referee #3**

**Specific comments**
*The setup of idealized models is always highly simplified. This simplification, however, must have a realistic reference. Otherwise, the idealized model would be just a toy. This study designed a model domain that consists of a tidal inlet and a semi-circle coastal sea. Does this domain represent the typical region around the British Isles where TEDs will be deployed? Is the setup of TEDs in the inlet channel the standard design of future TED deployment? If they are not, the representativeness of this idealized numerical model is less meaningful.*
The reviewer makes some valid points and we believe that our previous responses to Reviewer #1 and Reviewer #2 also address this comment. As we mention already, the domain is based on Strangford Lough, the location of the worlds first grid compliant TED, there are many regions where strong currents flow through narrow channels and will add this information in the manuscript.

*As idealized models are idealized, there is no need to validate the model results. Nonetheless, the model results should "look" generally reasonable. The simulated phytoplankton concentration is almost zero during the whole year in addition to a spring bloom occurred in April. Is this a typical phytoplankton cycle around the British Isles?*

The values used in the simulations were taken from measured values in the Irish Sea, however to ensure that everything is within plausible ecological bounds, we will check this.

*Specially, what is the atmospheric forcing of the model? What are the seasonal variations of the PAR scenarios? Did the model simulate the variation of water temperature that is important to phytoplankton growth?*

By addressing the earlier concerns regarding the PAR scenarios, we believe we will address this question also. The model does not explicitly simulate changes in water temperature but we consider that the results of our study stand without the explicit simulation of these temperature changes.

*The main founding of this study is that the deployment of TEDs reduces the phytoplankton concentration on basis of the comparison of the model results with/without TEDs. Certainly, this comparison is useful, but looks too superficial. Of importance is to explore the key processes (either physical or biological) through which the deployment of TEDs alters the phytoplankton dynamics. I feel that this manuscript would be more valuable if in-depth analysis of those processes were presented.*

As we state '*the paper addresses the applicability of using coupled hydrodynamic and biogeochemical models to investigate near-field (< 1 km), far-field (1–10 km) and regional scale (up to 30 km) effects of an array of tidal turbines on phytoplankton dynamics in a near coastal, shallow environment. In particular we use high resolution, 2-dimensional coupled hydrodynamic and biogeochemical NPZD model with two scenarios: no TEDs and an extreme, unrealistic setup with 55 TEDs, in an idealised domain*'. We believe that what we present, and with the addition of the constructive comments and suggestions from all three reviewers on light and nutrient effects/controls and how these may be effected, that this manuscript can be considered a foundation paper. We consider that this manuscript does this effectively, but accept that future manuscripts on this topic may explore, or build on, these results further, with additional analyses as required.